

# Environmental and vegetation control on the active layer and soil temperature in an Arctic tundra ecosystem in Alaska

Kevin J. Gonzalez Martinez[1,3], Donatella Zona[1], Trent Biggs[2], Kristine Bernabe[1], Danielle Sirivat[1,4], Francia Tenorio[1,3,4], Walter Oechel[1]

[1]Department Biology, San Diego State University, San Diego, CA 92182, USA

[2]Department Geography, San Diego State University, San Diego, CA 92182, USA

[3]NOAA EPP/MSI Center for Earth System Sciences and Remote Sensing Technologies II Fellow

[4]Department of Land, Air and Water Resources, University of California at Davis, United States

*Correspondence to*: Kevin J. Gonzalez (kevinjgonzalezmartinez@gmail.com) & dzona@sdsu.edu

**Abstract.** Permafrost soils contain approximately twice the amount of carbon than the atmosphere, which could be released as global warming continues. Increasing global temperatures have in fact the potential to result in increased permafrost degradation, and carbon loss into the atmosphere. To properly understand the potential release of the carbon stored in permafrost soils, it is critical to understand the environmental and vegetation control on the development of active layer, the upper soil layer that thaw during the growing season in the Arctic. Arctic tundra ecosystems are dominated by mosses, which compose approximately 40% of the vegetation, and have a critical role in regulating the heat condition into the soil. Given their importance, the role that mosses play on permafrost degradation should be investigated in more details. This study measured soil temperature together with thaw depth, a range of environmental variables, and moss thickness, to identify the most important controls on the development of the active layer across 124 plots in continuous permafrost tundra ecosystems. We found that a thicker moss layer insulated the soil and resulted in cooler temperatures deeper in the soil, despite warmer surface temperatures. A thicker moss layer was associated with a deeper depth of thaw, likely for the higher growth of mosses in the drier and warmer topographically higher elevation areas. The protective role of mosses was only relevant for the first ~3



cm of the green moss layer, suggesting that the living moss layer was more important in regulating soil temperature, possibly
through a higher ability to retain water. Soil moisture was in fact an important control on surface and deeper soil temperatures,
with wetter soils been associated with cooler surface temperatures because of the higher evaporative cooling, and warmer
deeper temperatures likely because of the larger heat conduction to deeper soils. Overall, this study highlights the importance
of a green living moss layer on soil temperature and thaw depth. Mosses are among the most vulnerable vegetation to
hydrological changes, given their lack of a rooting system, and their sensitivity to climate change should be considered when
predicting the response of permafrost thaw to climate change.
**Short summary**
Permafrost soils contain twice the amount of carbon than the atmosphere, and its release could majorly affect global
temperatures. This study found that a thicker moss layer resulted in cooler temperatures deeper in the soil, despite warmer
surface temperatures. The top green living moss layer was the most important in regulating the soil temperatures and should
be considered when predicting the response of permafrost thaw to climate change.

**1 Introduction**
Northern high-latitude ecosystems are characterized by the presence of permafrost, a soil layer that remains frozen for at least
two consecutive years (Permafrost Subcommittee 1988). The active layer (also known as the thaw depth) is the layer of soil
of variable depth found directly above the permafrost which only thaws during the summer (Permafrost Subcommittee 1988).
The depth of this active layer depends on a combination of factors, including abiotic factors such as air temperatures, solar
radiation, and soil moisture, which may increase subsurface heat and associated belowground thermal difference (Dafflon et
al. 2017; Fisher et al. 2016; Permafrost Subcommittee 1988; Schuur et al. 2015; Williams et al. 2020). Vegetation also has a
role in the development of the active layer by potentially decreasing heat transfer below the surface by serving as a layer of
insulation when dry or increasing heat transfer when wet (Beringer et al. 2005; Blok et al. 2010; Blok et al. 2011; Hayashi et
al. 2007; Hrbáček et al. 2020; Park et al. 2018; Porada et al. 2016). Globally, permafrost and the active layer currently store





an estimated 15 gigatons of carbon (Koven et al. 2011; Schurr 2019). The increasing severity of climate change currently
threatens the stability of this sequestered carbon (Miner et al. 2022; Schaphoff et al. 2013; Schuur et al. 2008). The release of
the vast carbon stored in these high-latitude soils can potentially affect the climate at the global scale through positive feedback
loops in which increased atmospheric carbon may further greenhouse emissions and permafrost degradation (Davidson and
Janssens 2006; Schuur et al. 2008). These positive feedback loops further contribute to the rapid change in climate in the
northern high-latitude ecosystems known as "Arctic amplification" (Dai et al. 2019).
Given the large carbon store of permafrost soils (Hugelius et al. 2014), it is critical to improve understanding of the
environmental and vegetation controls on active layer development. Mosses compromise approximately 40% of the Arctic
sedge tundra and exist in "mats" alongside other forms of vegetation directly above permafrost soils and their associated ice
wedges (Euskirchen et al. 2009). Mosses play a significant role in regulating thaw depths by reducing the heat penetration
belowground when dry or increasing it when wet (Beringer et al. 2005; Blok et al. 2010; Blok et al. 2011; Hayashi et al. 2007;
Hrbáček et al. 2020; Park et al. 2018). Mosses retain a unique adaptation to desiccation that allows them to desiccate while
remaining alive and to create a layer of tissue, which thermally insulates soils (Blok et al. 2011; Park et al. 2018). The dried
mats have an insulation effect with increased moss community coverage and layer thickness associated with lower soil
temperatures in deeper soil layers and a shallower active layer depth (Blok et al. 2011; Park et al. 2018; van der Wal et al.
2001). A wetter moss layer should result in increased heat conduction in the soil and a deeper active layer depth (Hayashi et
al. 2007; Hrbáček et al. 2020; Park et al. 2018).
The overall control that mosses and other environmental drivers have on permafrost degradation is still not fully understood
(Fisher et al. 2018; Luo et al. 2016). The Arctic is experiencing warmer temperatures and increased precipitation between
August to October (Boisvert and Strove 2015, Fujinami et al. 2016). The resulting increase in moisture availability can increase
thermal conduction in deeper soil layers in turn supporting thawing of the soil (Beringer et al. 2005; Blok et al. 2011; Hayashi
et al. 2007; Hrbáček et al. 2020; Park et al. 2018). This increase in thermal conduction may lead to increased permafrost soil
thawing which in turn would lead to increased releases of greenhouse gas emissions (Miner et al. 2021; Schaefer et al. 2014;





van Huissteden and Dolman 2012). Therefore, an increased depth of thaw can create a positive feedback loop which can further
exacerbate the effects of climate change in the Arctic.
To further our understanding of the controls on the development of the active layer, a range of environmental drivers and
vegetation characteristics should be investigated, including soil moisture, solar radiation, and air and soil temperatures together
with the thickness of the moss mat across the fine scale, micro topographically variable polygonal tundra ecosystems. The goal
of this research is to identify the biotic and abiotic controls regulating soil temperature and the thawing of the active layer
across a range of microtopographic areas in a wet sedge arctic tundra ecosystem in Alaska. We expect thicker mosses to be
associated with shallower depth of thaw and decreased belowground temperatures. We anticipate moss patches with increased
soil water content to be associated with deeper depth of thaw as well as increased belowground temperatures given an increase
in heat conduction with higher soil moisture.
**2 Methods**
**2.1 Study Sites**
The sites of this research are near Utqiaġvik, Alaska (formerly known as Barrow), which is the largest town in the Alaskan
North Slope Borough. Our team has been maintaining several micrometeorological and eddy covariance towers in Utqiaġvik
over the last decades (Zona et al., 2016). This research was conducted near two of these eddy covariance towers, which were
in operation from 2005-2009 (Fig. 1b, the US-Ben (North), and US-Bec (Central) sites, established during the Biocomplexity
Experiment, see Zona et al., 2009; Zona et al., 2012). Environmental drivers (such as air temperature and photosynthetic active
radiation (PAR)) measured from another tower still operational since 2005 (US-Bes), in close proximity to the US-Ben and
US-Bes sites, were also included in this study. The locations of the US-Bes, US-Bec, and US-Ben are: 71.2809N, -156.5965W;
71.28316N, -156.60342W; and 71.28628N, -156.60424W respectively (Zona et al., 2009). These sites are located in a drain
lake basin ecosystem with vegetation classified as wet sedges tundra  dominated by mosses, lichens, and graminoids with
patches of water and partially to fully submerged patches of vegetation (Davidson et al., 2016). Given the proximity of these
sites (US-Ben and US-Bec being within 662-meters and 356-meters of US-Bes respectively), we assume that the air





temperature and PAR collected in US-Bes were representative of the US-Ben and US-Bec sites. Access to US-Bes, US-Bec,
and US-Ben sites was facilitated by the establishment of boardwalks during the Biocomplexity experiment in summer 2005
(Zona et al., 2009). These boardwalks allowed sampling across the sites while limiting disturbance. Data was collected every
two meters in both US-Ben and US-Bec across 124-meters transects that parallel historical water table data collection (Zona
et al., 2012), for a total of 62 plots in US-Ben and 62 in US-Bec.

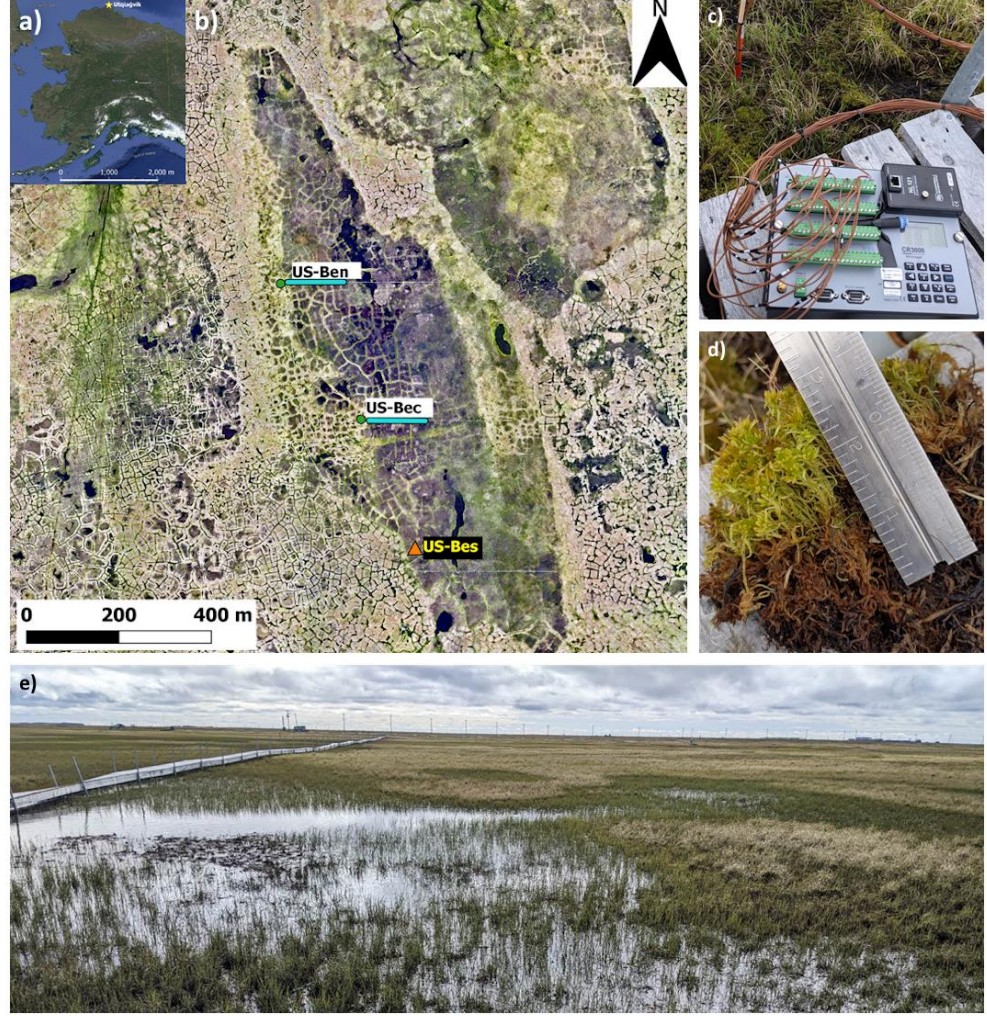


**Figure 1: Study site: (a) Map of Alaska showing the general location of Utqiaġvik accessed via ©Google Earth Pro**
**Copernicus/Landsat satellite imagery (2023). (b) A high-resolution orthorectified camera imagery mosaic acquired by the National**





**Ecological Observatory Network (NEON) Airborne Observation Platform (AOP) highlighting the US-Ben and US-Bec transects (with the transect lengths highlighted in blue) and the US-Bes tower site. (c) The CR3000 datalogger with 21 thermocouples for belowground and air temperature recordings every centimetre. (d) A cross section of the moss layer showing the total (lowest brown point to surface) moss layer thickness and green (living) moss layer thickness. (e) A photograph of the research site, showing the boardwalk used for sampling.**

## 2.2 Vegetation and environmental controls

The data collection was performed at 2-meter intervals along two 124-meter transects at US-Bec and US-Ben (Fig. 1b). In each of these plots we recorded the thickness of the total moss layer and the thickness of the green living moss layer. A previous in-depth vegetation analysis from our team showed that the general region where this data was collected was composed primarily of graminoids as the dominant vascular plant group, and by mosses such as *Sphagnum sp*. and *Drepanocladus sp.* (Davidson et al., 2016). We also collected data on the dominant moss genus at every point and found *Sphagnum sp.* and *Drepanocladus sp.* to be the dominant genera at our sites with most of the plots being *Sphagnum sp.* (N = 55) or some combination of predominantly *Sphagnum sp.* and *Drepanocladus sp.* (N = 46) and a limited number of plots with only *Drepanocladus sp.* (N = 21). The moss layer thickness and genus identification were recorded in sections of approximately 25-square cm (5 cm x 5 cm) in each of the 2-meter plots only once (i.e., the first week of July in 2021 for all plots and a subset of these plots on the first week of July in 2022) during the peak season (i.e., the first week of July to the second week of August 2021) to reduce the disturbance to the moss layer. These sections were carefully removed using a serrated knife trying to limit damage to surrounding vegetation. Afterwards, the thickness of the moss mat was measured with a ruler (Fig. 1d), and the samples were reinserted into the sampling locations.

In each of the plots for each of the sites (N=62 US-Ben, and N=62 for US-Bec) we also recorded moss and soil temperature every cm from 1 cm below the surface until 20 cm below ground on a weekly basis using type T thermocouples connected to a CR3000 datalogger (Campbell Scientific, Logan, UT, USA). These temperature profiles allowed us to determine how the presence and thickness of the moss layer affected the thermal difference across the moss and soil layers. These 21





thermocouples were attached to a fiberglass probe, which facilitated insertion in the moss layer and soil. Each point was
measured for approximately 3 minutes as the temperature readings stabilized within the first couple of minutes. We also
collected soil water content (percent water) weekly in the first 5-cm of the moss or soil layer using a FieldScout TDR300
(Spectrum Technologies, Aurora, IL, USA) and 5-cm rods (Beringer et al., 2005; Hayashi et al., 2007; Hrbáček et al., 2020).
The FieldScout was calibrated using local water samples to account for nutrients which may influence conductivity. Thaw
depth and water table levels were also collected weekly in each of the sampling plots using a metal and wooden probe
respectively with markings indicating intervals of 1-cm depths. Water table measurements were collected inside PVC pipes
(with holes every 1 cm) previously installed along the transects (Zona et al., 2009; Zona et al., 2012) in each of the sampling
locations. This data collection was repeated for a second field season in a subset of the plots (N = 20 in Summer 2022 vs. N =
124 in Summer 2021) to reduce disturbance to the sites; samples collected in these different years were compared, showing
good agreement between the measured moss thickness in the 2021 and 2022 field seasons ($R^2$ = 91.82%, p-value < 0.001, 2022
Moss Thickness (cm) = 1.01967 * 2021 Moss Thickness (cm) + 0.228272).
Environmental variables collected by the eddy covariance US-Bes tower, included PAR, air temperature, local surface and
subsurface soil temperature, relative humidity, wind speed, and net radiation. Elevation above sea level was collected in each
of the sampling plots at US-Ben and US-Bec with a dGPS as reported in Zona et al. (2012). These measurements described
the microtopography of each of the sampling plots and allowed us to test the role on microtopography on the environmental
conditions, vegetation, and active layer development.

**3 Statistical Analysis**
All statistical analyses were conducted using R version 4.2.0 (R Core Team, 2022); the caret, leaps, and MASS packages
(Kuhn, 2022; Lumley and Miller, 2022; Posit Team, 2022; R Core Team, 2022; Venables and Ripley, 2002), and stepwise
multiple regressions and univariate regressions (both linear and nonlinear) were used to test for and model the relationships
between the deepest (minimum given the negative sign) thaw depth ($D_{thaw}$) or the temperature difference between 1 and 15 cm





belowground ($dT_{1\_15}$) and environmental predictors, and to evaluate the collinearity in predictors. For the stepwise multiple
regressions, we tested the correlations between predictor variables, given that covarying variables can create issue when
included together in the same model; variables that met cutoff points ($-0.65 \geq r \geq 0.65$ and p-value $< 0.05$) were not included
together in the stepwise multiple regression models. Data from both 2021 and 2022 were aggregated by both site and distance
along each transect. The predictor variables included median, maximum, and minimum values of water table level, soil water
content, temperature at 15 cm belowground, 10 cm belowground, 5 cm belowground, 1 cm belowground, 5 cm aboveground,
green moss layer thickness, total moss layer thickness, and PAR. Deviation from the mean elevation (dz) was calculated as a
simple difference between each elevation point and the mean elevation. $dT_{1\_15}$ was calculated as the difference in temperatures
recorded at 1 cm belowground and 15 cm belowground; -15 cm was selected as it was the thaw depth most consistently
represented at most of the sites during the entire collection period. We used the median $dT_{1\_15}$ for each plot along each transect
for both 2021 and 2022, given that only one elevation value was recorded for each plot. We tested the influence of moss genus
on the statistical analysis but did not find it significant, so we did not include it in the results.
$dT_{1\_15}$ was modelled as a function of the median air temperature, dz, the soil water content, and both the midsummer green
and total moss layer thicknesses. We tested the relationship between $D_{thaw}$ as a function of dz, median water table level, median
soil water content, maximum soil temperatures, median local air temperatures, 2021 midsummer green and total moss layer
thicknesses, median PAR, and $dT_{1\_15}$. $dT_{1\_15}$ was regressed against the median water table level, median soil water content,
$D_{thaw}$, median local air temperatures, green and total moss layer thicknesses, median PAR, and dz. We tested the collinearity
of predictor variables; covarying variables ($-0.65 \geq r \geq 0.65$ and p-value $< 0.05$) were not included together in the stepwise
regression models, and each variable was also tested separately in a univariate model to rank their relative importance as
explanatory variable.
To evaluate the environmental controls on either thaw depth, and soil temperature (at -1 cm and -15 cm depth), we tested a
variety of non-linear regressions, including different order polynomial and logarithmic regressions, given that ecological
processes can have complex relationships (Zona et al., 2023). However, when evaluating these models, we did not find an
ecological explanation for the statistical models with the highest explanatory power which were at the time third order





polynomial models. Therefore, we applied piecewise regressions to test the occurrence of a breakpoint in the linear regressions,
using the segmented package in R (Muggeo, 2003; 2008). When the occurrence of a breakpoint was significant, we included
separate linear regressions for the two datasets separated by the breakpoint.

**4 Results**

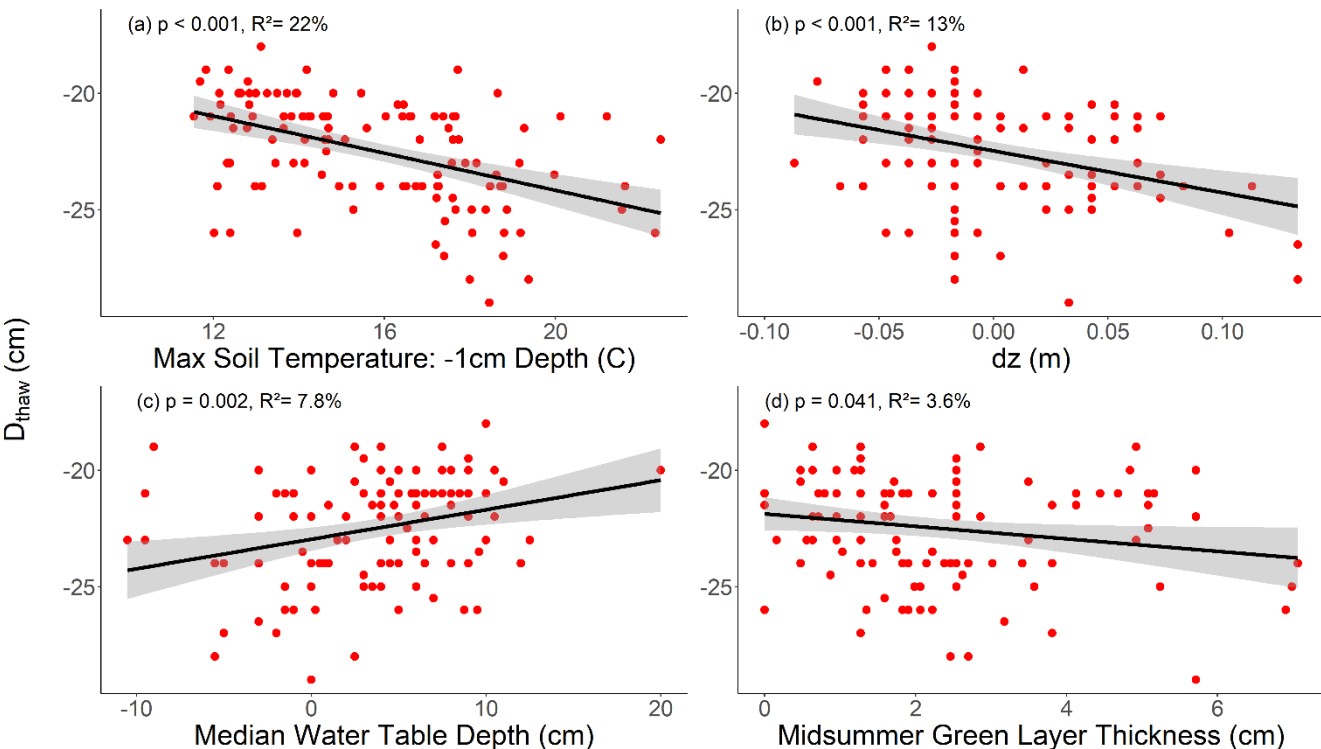

**Figure 2: Relationship between the deepest thaw depth ($D_{thaw}$) and the indicated variables for the entire study period (summer 2021**
**and 2022) and for both study sites combined, demonstrating  a) a negative relationship between the maximum soil temperature at -**
**1 cm depths and deepest thaw depth, b) a negative relationship between the deviation from the mean elevation (dz) and the deepest**
**thaw depth, c) a positive relationship between water table level and the deepest thaw depth, and d) a negative relationship between**
**green moss layer thickness during the middle of the 2021 summer and the deepest thaw depth.**



Several environmental variables were strongly correlated (Table S1). The strongly correlated variables ($-0.65 \geq r \geq 0.65$ and
$p$-value $< 0.05$) were not included together in the regression models of $D_{thaw}$, $dT_{1\_15}$, and soil temperature at -1 and -15 cm. See
Table S1 for details on the correlation coefficients and $p$-values among those variables.
The maximum temperature for each measurement point aggregated across both years at -1 cm belowground had the highest
explanatory power in the stepwise model of the deepest thaw depth for each measurement point aggregated across both years
(p-value $< 0.001$, $R^2 = 22\%$, Akaike information criterion (AIC) = 500, Table 1). A similar result was obtained from the
comparison of univariate regressions, where the maximum soil temperature at -1 cm had the highest explanatory power on the
variability of thaw depth. The stepwise model for $dT_{1\_15}$ identified the median soil water content, deepest thaw depth, and dz
as the only significant variables among the variables tested (i.e., median soil water content, the deepest thaw depth, median air
temperature, total moss layer thickness, and the dz, p-value $< 0.001$, $R^2 = 45\%$, AIC = 525, Table 2). A complete list of the
statistics of the univariate models are included in Tables 3 and 4.
**Table 1: Statistics of the stepwise linear regression model of the deepest thaw depth (cm) for summer 2021 and 2022 across all the**
**plots, which selected the maximum soil temperature at -1 cm depth as the variable with the highest explanatory power and the only**
**significant predictor. Included are the regression coefficient, $R^2$, p-value, the Akaike information criterion (AIC).**

| | Deepest Thaw Depth | | | |
|---|---|---|---|---|
| **Variable** | **Coefficient** | **$R^2$** | ***p*-value** | **AIC** |
| Maximum Soil Temperature – -1 cm | -0.40 | 22% | < 0.001 | 500 |


**Table 2: Statistics of the stepwise linear regression model of the temperature difference between the 1 and 15 cm depths ($dT_{1\_15}$)**
**which selected the following variables as those with the highest explanatory power. Included are the coefficients, $R^2$, p-values, and**
**the Akaike information criterion (AIC). The overall $p$-value for the entire model was < 0.001.**





| Temperature Difference | | | | |
|---|---|---|---|---|
| **Variable** | **Coefficient** | **$R^2$** | ***p*-value** | **AIC** |
| Median Soil Water Content | -0.09 | | 0.003 | |
| Deepest Thaw Depth | -0.33 | 45% | 0.001 | 525 |
| Deviation from the Mean Elevation (dz) | 27.94 | | < 0.001 | |


After ranking and comparing all models including the variables listed in Table S1, maximum soil temperatures at -1 cm, dz,
median water table level, and green moss layer thickness explained the largest percentage of variation in the deepest thaw
depth (Fig. 2). Soil temperature alone explained about 22% of the variability in the deepest thaw depth (Table 1). Similarly,
$dT_{1\_15}$ was mostly explained by the dz, soil water content, and green and total moss layer thicknesses (Fig. 3). The dz had the
highest explanatory power explaining 34% of the variability in $dT_{1\_15}$ (Fig. 3), and the addition of soil water content and the
deeper thaw depth increased the explanatory power to 45% (Table 2).

**Table 3: Statistics of the univariate simple linear regressions of the deepest thaw depth and the indicated variables including the**
**entire dataset. The variables are ranked based on the simple linear model's $R^2$ value. Included are the $R^2$, p-value, the Akaike**
**information criterion (AIC), and the breakpoint if statistically significant.**

| Deepest Thaw Depth | | | | |
|---|---|---|---|---|
| **Predictor Variable** | **Breakpoint** | **$R^2$** | **p-value** | **AIC** |



| | | | | |
|---|---|---|---|---|
| **Maximum Soil Temperature – -1 cm** | N/A | Linear: 22% | Linear: < 0.001 | Linear: 500 |
| **Belowground Temperature Difference** | N/A | Linear: 19% | Linear: < 0.001 | Linear: 504 |
| **Median PAR** | N/A | Linear: 13% | Linear: <0.001 | Linear: 512 |
| **Difference from Mean Elevation** | N/A | Linear: 13% | Linear: < 0.001 | Linear: 513 |
| **Median Water Table Level** | N/A | Linear: 7.8% | Linear: 0.002 | Linear: 519 |
| **Maximum Soil Temperature – 5 cm** | N/A | Linear: 7.1% | Linear: 0.004 | Linear: 520 |
| **Green Moss Layer Thickness** | N/A | Linear: 3.6% | Linear: 0.041 | Linear: 524 |


**Table 4. Statistics of the univariate simple linear and piecewise regressions of the temperature difference ($dT_{1\_15}$) and the indicated**
**variables including the entire dataset. Included are the $R^2$, p-value, the Akaike information criterion (AIC), and the breakpoint if**
**statistically significant. The significance in the difference between models is included in the column "ANOVA" only when the p-**
**value was less than 0.05 (e.g., the significance in the difference between the linear (L) and the piecewise (PW) was 0.005 for the**
**median local air temperature). The best models are highlighted in bold.**

| Thermal Difference |
|---|



| Variable | Breakpoint | R² | p-value | AIC | ANOVA |
|---|---|---|---|---|---|
| **Median PAR** | N/A | **Linear: 43%** | **Linear: < 0.001** | **Linear: 523** | |
| **Median Local Air Temperature** | x = 12.6 | Linear: 41% | Linear: < 0.001 | Linear: 528 | |
| | | **x < 12.6: 12%** | **x < 12.6: < 0.001** | **x < 12.6: 539** | **L-PW: 0.005** |
| | | **x > 12.6: 50%** | **x > 12.6: 0.049** | **x > 12.6: 36.7** | |
| **Median Water Table Level** | x = -1.1 | Linear: 40% | Linear: < 0.001 | Linear: 530 | |
| | | x < -1.1: 6.3% | x < -1.1: 0.330 | x < -1.1: 67.3 | **L-PW: < 0.001** |
| | | **x > -1.1: 42%** | **x > -1.1: < 0.001** | **x > -1.1: 448** | |
| **Difference from Mean Elevation** | N/A | **Linear: 34%** | **Linear: < 0.001** | **Linear: 541** | |
| **Minimum (Deepest) Thaw Depth** | N/A | **Linear: 19%** | **Linear: < 0.001** | **Linear: 566** | |
| **Median Soil Water Content** | x = 74.4 | Linear: 15% | Linear: < 0.001 | Linear: 570 | |
| | | x < 74.4: 0.36% | x < 74.4: 0.725 | x < 74.4: 167 | **L-PW: < 0.001** |
| | | **x > 74.4: 19%** | **x > 74.4: < 0.001** | **x > 74.4: 381** | |
| **Green Moss Layer Thickness** | x = 3.1 | Linear: 14% | Linear: < 0.001 | Linear: 572 | |
| | | **x < 3.1: 21%** | **x < 3.1: < 0.001** | **x < 3.1: 439** | **L-PW: < 0.001** |



| | | x > 3.1: 4.6% | x > 3.1: 0.294 | x > 3.1: 125 |
|---|---|---|---|---|
| **Total Moss Layer Thickness** | N/A | **Linear: 7.0%** | **Linear: < 0.001** | **Linear: 581** |


We found a negative relationship between the green moss layer thickness and deepest thaw depth (Fig. 2d, *p*-value = 0.041,
$R^2$ = 3.6%) and a positive relationship between the green (Fig. 3c, *p*-value < 0.001, $R^2$ = 21% when the green layer thickness
was below 3.1 cm) or total layer thickness (Fig. 3d, *p*-value < 0.001, $R^2$ = 7.0%) and $dT_{1\_15}$. A positive relationship was
observed between green (Fig. 4b, *p*-value < 0.001, $R^2$ = 9.7%) or total layer thickness (Fig. 4a, *p*-value = 0.031, $R^2$ = 4.0%)
and the maximum soil temperature at the surface, but a negative relationship between increasing green (Fig. 4e, *p*-value =
0.002, $R^2$ = 7.9%) (or total layer thickness, Fig. 4d, *p*-value = 0.006, $R^2$ = 6.3%) and the maximum soil temperature at -15 cm
depths. We observed a negative relationship between the deepest thaw depth and dz (Fig. 2b, *p*-value < 0.001, $R^2$ = 13%) and
a positive relationship between $dT_{1\_15}$ and dz (Fig. 3a, *p*-value < 0.001, $R^2$ = 34%). We noticed a negative relationship between
soil temperatures at -1 cm depths and deepest thaw depth (Fig. 2a, *p*-value < 0.001, $R^2$ = 22%). We observed a positive
relationship between increasing green moss layer thickness (Fig. 6b, *p*-value < 0.001, $R^2$ = 13%), or total moss layer thickness
(Fig. 6a, *p*-value < 0.001, $R^2$ = 15%), and dz.

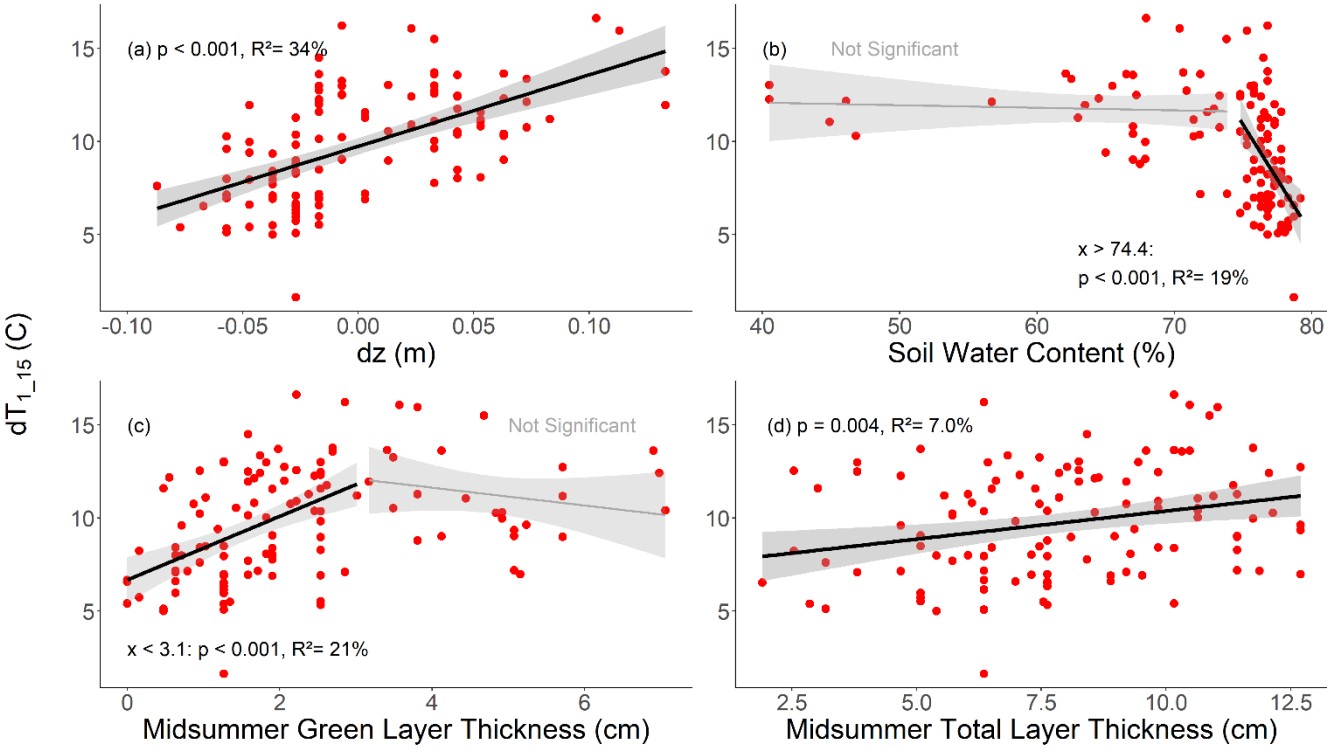

**Figure 3: Relationship between the temperature difference (between soil temperature at -1 cm and at -15 cm belowground) and the indicated variables for the entire sampling period (summer 2021 and 2022) across all the sampling plots, showing a) a positive relationship between the deviation from the mean elevation (dz) and the temperature difference, b) a negative relationship between soil water content and the temperature difference when soil water content is greater than 74.4%, c) a positive relationship between the green moss layer thickness during the middle of the 2021 summer and the temperature difference when the green moss layer thickness was less than 3.1 cm, and d) a positive relationship between the total moss layer thickness and the temperature difference.**

Soil water content (soil water content) had a negative relationship with temperature difference when soil water content was greater than 74.4% (Fig. 3b, $p$-value < 0.001, $R^2$ = 19%) and no significant relationship below that threshold. While soil temperatures at -1 cm were negatively associated with soil water content past a threshold of 74.3% (no significant relationship was noted prior to this threshold), the temperature deeper in the soil (15 cm below ground) was positively correlated with soil water content past a soil water content of 76.4% (Fig. 4). A positive relationship was seen between water table level and deepest thaw depth (Fig. 2c, $p$-value = 0.002, $R^2$ = 7.8%).

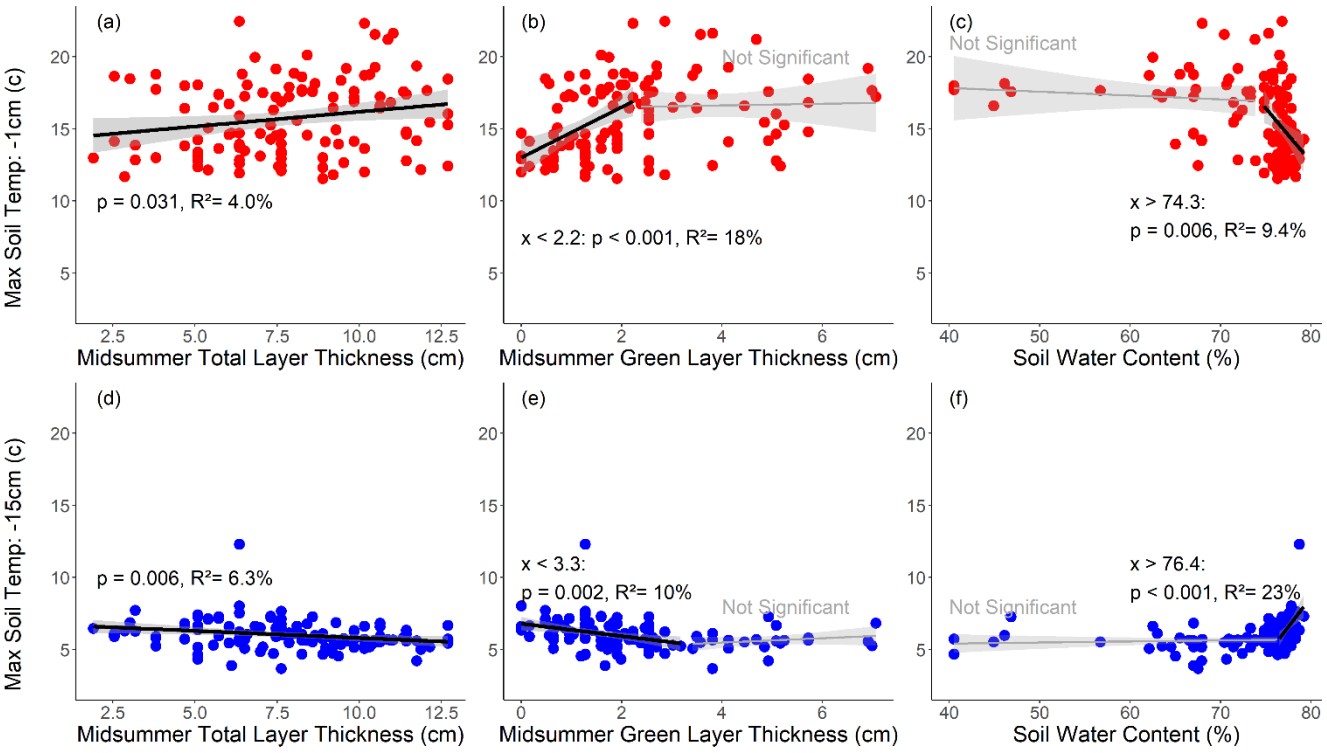

**Figure 4: Relationships between the maximum soil temperature and the indicated variables for the entire periods of measurements aggregated (summer 2021 and 2022), and for all the sampling plots. While green and total moss layer thicknesses was positively associated with higher temperatures closer to the surface (a and b), thicker moss layers were associated with cooler temperatures at deeper depths (d and e). As the percent soil water content increased, the superficial temperatures decreased (c) but temperature in deeper soil layers increased (f).**

Given the significant relationship between soil temperature and relative elevation, we conducted a two-sample t-test to compare surface temperatures at -1 cm depth across both transects in areas above and below the mean elevation. There was a significant difference in temperature between areas above the mean elevation (4.2 cm) and below the mean elevation. The median temperature of higher elevation areas was (M = 17.3 ± 4.3 °C) and those below the mean elevation (M = 14.8 ± 4.9 °C); t(107.85) = 5.8, p < 0.001, Fig. 5. Another two-sample t-test showed percent soil water content was lower in areas with elevations above the mean elevation (M = 70.6 ± 17.4%) compared with areas below the mean elevation (M = 74.7 ± 12.8%) ($p$ = 0.007), as shown in Fig. 5, and summarized in Fig. 7.





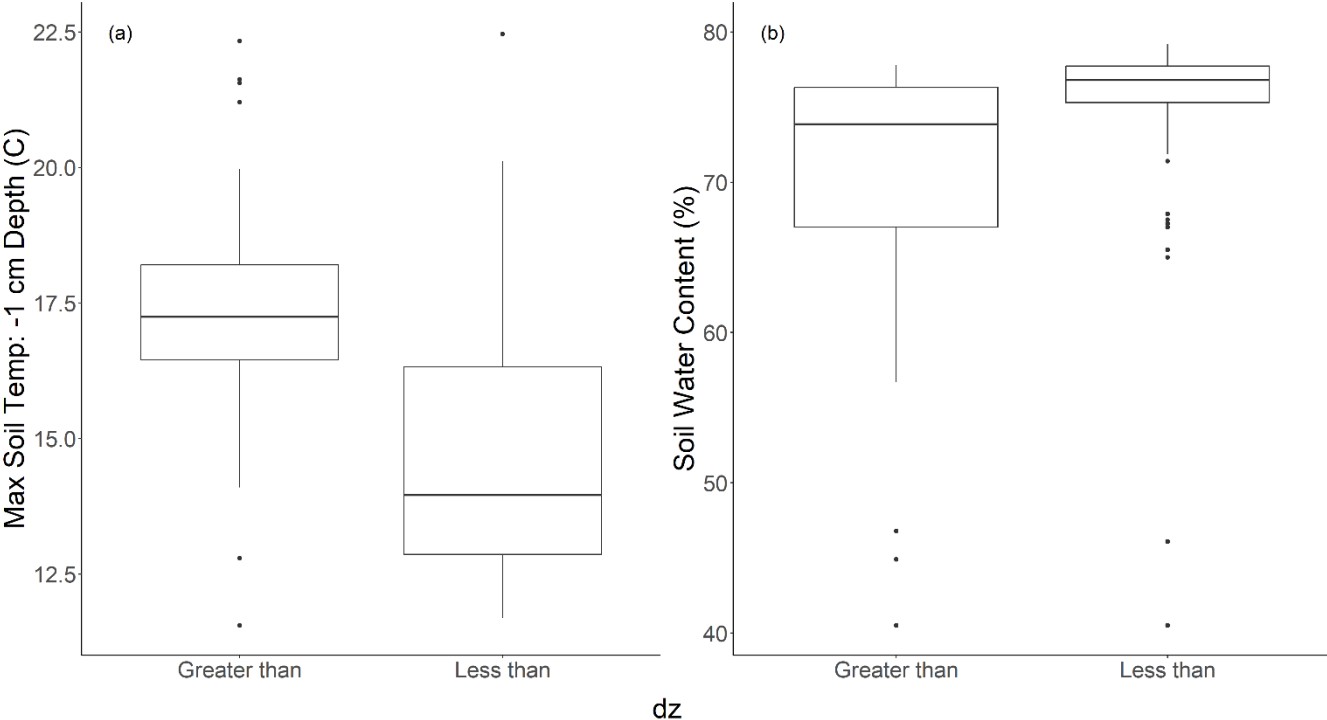

**Figure 5: Comparison of (a) maximum soil temperature ranges at a -1 cm depth and (b) soil water content based on the deviation from the mean elevation (dz) grouped by areas greater than and less than the mean elevation for the entire periods of measurements (summer 2021 and 2022), and for all the sampling plots. a) Areas with higher elevations had a higher observed average soil temperature. b) Areas with lower observed average soil water content.**





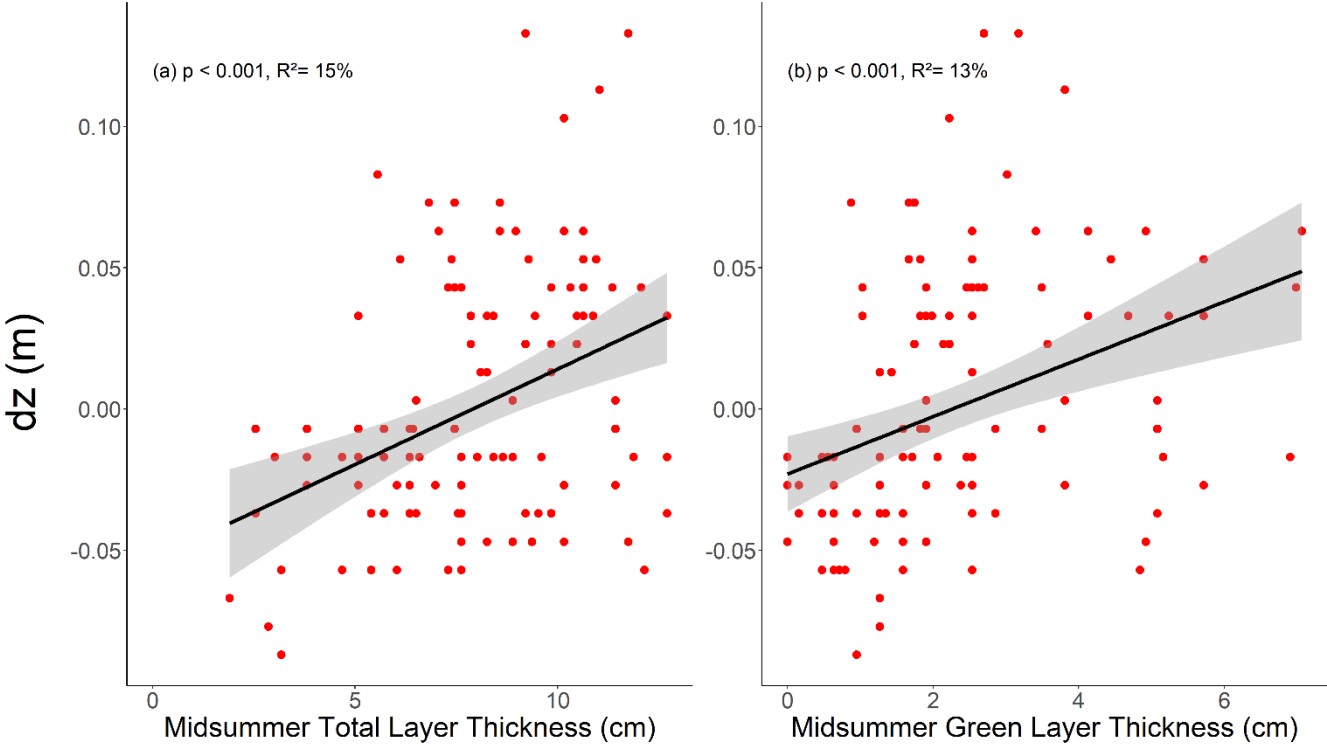

**Figure 6: Relationships between (a) increasing midsummer (2021) total layer thickness or (b) midsummer (2021) green layer thickness and the deviation from the mean elevation (dz) suggesting the occurrence of a thicker moss layer in higher topographic areas.**





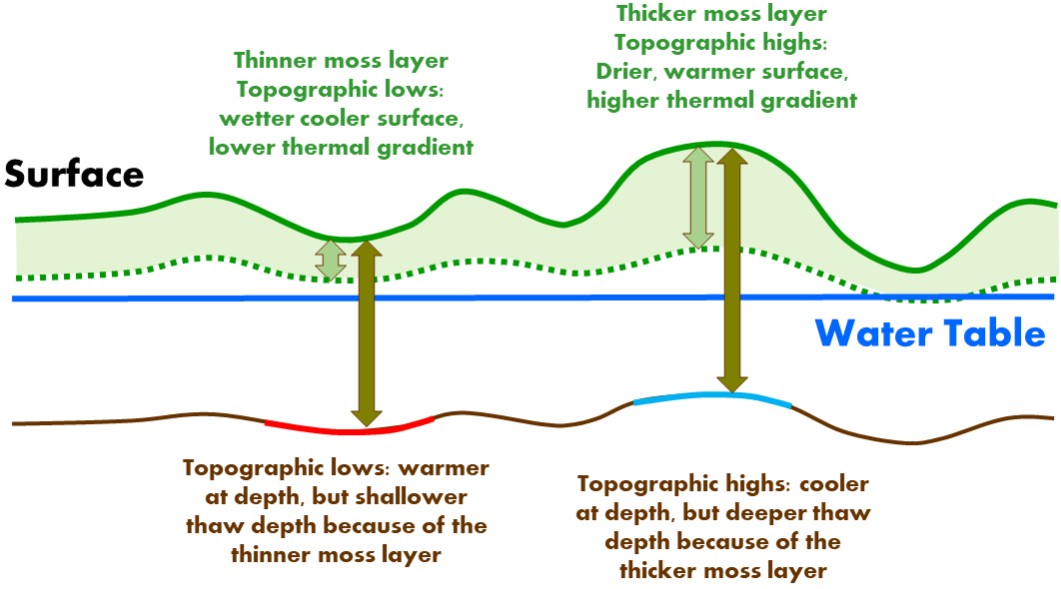


**Figure 7: A visual summary of the relationships highlighted by the regression analyses highlighting how thinner moss layers have a seemingly shallower depth of thaw and lower surface but higher belowground temperatures as opposed to thicker moss layers that have higher surface temperatures and seemingly deeper depth of thaw but lower belowground temperatures.**


**5 Discussion**

We found that moss layer thickness, microtopography, and soil moisture exerted important controls on soil temperature
profiles. Microtopographic lower elevation areas had higher soil water content and lower temperature gradients than
topographic highs. Thicker green moss layers co-occurred with cooler deep (-15 cm below the surface) soil temperature,
which is consistent with a higher thermal insulation (Beringer et al., 2005; Fisher et al., 2016; Hrbáček et al., 2020;
Soudzilovskaia et al., 2017). Thicker mosses insulate the soil (Blok et al., 2011; Chadburn et al., 2015; Heijmans et al., 2022;
Hrbáček et al., 2020; Park et al., 2018) as shown by the cooler deeper soil temperature, even with warmer near surface (i.e. -
1 cm) temperatures. Moss thickness was higher in higher elevation areas possibly because their higher surface temperature





might have stimulated moss growth, in these temperature limited ecosystems (Harley et al., 1989; Bengtsson et al., 2021).
Near surface temperature was the main control on thaw depth, consistent with previous studies (Dafflon et al., 2017; Schuur
et al., 2015). Soil moisture was also a very relevant control on soil temperature, with opposite relationships for the near
surface and deeper soil temperatures. Cooler near surface temperature co-occurred with higher soil moisture and could be
explained by the higher evaporative cooling of a wetter moss layer (Heijmans et al., 2004a, b) and by the higher thermal
conductivity and rates of heat transfer to lower soil layers. The higher thermal conductivity and heat penetration in wetter
soils could also explain the higher soil temperature in deeper soil layers (-15 cm) (Soudzilovskaia et al., 2013; Fisher et al.,
2016; Hrbáček et al., 2020; Soudzilovskaia et al., 2017; Curasi et al., 2016; Hinkel and Nelson, 2003; Hinkel et al., 2001;
Shiklomanov et al., 2010).
We observed a significant breakpoint, and different linear regressions for the shallower (~2-3 cm) than deeper green (living)
moss layers in describing the relationship between green moss layer thickness and temperatures at -1 cm and -15 cm. This
would suggest that the top living layer of mosses has the most relevant role in regulating soil temperature. The larger
importance of the top moss layer in insulating the soil is consistent with the reported significant relationship between the top
~15-20 cm of the moss layer moss layer thickness and permafrost thaw (Douglass et al., 2008). The significance in the
breakpoint was only observed for the relationship between the green layer thickness and soil temperatures (at -1 cm and at -
15 cm depth), while no significant breakpoint was observed in the relationship between the total moss thickness and soil
temperatures. This is likely because of the higher water retention of the handlike structures (papillae) present in the green
moss layer (Clymo, 1970; Dykas, 2018), which could increase their importance in regulating the heat transfer. Similarly, the
lack of a significant response between temperatures at -1 and -15 cm and soil moisture until approximately 74.3% and 76.4%
respectively, is consistent with the observed exponential relationship between the thermal conductivity and relative moisture
content reported for bryophytes and lichen in permafrost ecosystems, with a steeper relationship at higher moisture levels
(Porada et al., 2016). Overall, the relationship between deviation from mean elevation (dz) and moss thickness together with
the relationships between moss thickness and -15 cm temperatures confirms that microtopography dominated ecosystem
functioning in these arctic tundra ecosystems (Zona et al., 2011; Wilkman et al., 2018).






## 6 Conclusion

The results of this study support the importance of mosses and soil moisture to insulate permafrost. A thicker moss layer is
associated with warmer near surface temperature but with cooler deeper soil temperature and larger thermal gradient because
of higher thermal insulation. Future studies should better define the role of moisture on heat penetration to deeper soil layers,
across a wider range of soil moisture, as this study was mostly focused on very wet ecosystems. Additionally, a wider range
of moss thickness together with the role of other vegetation types should be considered when modelling the soil temperature
and thaw depth to understand the controls on the integrity of permafrost.

## 7 Data Availability

Donatella Zona, & Kevin Gonzalez. (2023). *Environmental and vegetation control on active layer and soil temperature in an*
*Arctic tundra ecosystem in Alaska*. Arctic Data Center. urn:uuid:0e00e7c3-d2a1-4065-bba6-3c664b983990.

## 8 Author Contribution

DZ and WCO acquired funding to support the project leading to this publication. DZ conceptualized the project with KG
conducting the investigation process involving data collection. KG conducted formal analysis with contributions from DZ
and TB. KG prepared the manuscript with contributions and critical review from all co-authors. All co-authors provided
feedback on the data analysis and contributed to writing the manuscript.

## 9 Competing Interests



The authors declare that they have no conflict of interest.

**10 Acknowledgements**

This study was funded by the Office of Polar Programs of the National Science Foundation (NSF) and awarded to D.Z.,
W.C.O. (award numbers 2149988 and 1932900) with additional support by the ABoVE (80NSSC21K1350) Program. The
Alaskan sites are on land owned by the Ukpeaġvik Iñupiat Corporation (UIC) and are a part of Iñupiat indigenous
community. K.G. and F.T. are supported in part by the NOAA Educational Partnership Program/Minority-Serving
Institutions award NA22SEC4810016 Center for Earth System Sciences and Remote Sensing Technologies II. Contents are
solely the responsibility of the author(s) and do not represent the official views of NOAA or the U.S. Department of
Commerce.



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
