# Peer review of "Environmental and vegetation control on the active layer and soil"

_Biogeosciences, 2023_

## Author Comment (AC1)

**This manuscript describes a study on the controlling factors of soil temperature and thaw depth for arctic tundra in Alaska, with a focus on the role of mosses. The study collected data on thickness and composition of moss layer (green/total), soil temperature profiles, soil water content, water table, thaw depth along two transects in a well-established research site. The study found that moss layer thickness, microtopography and soil moisture influence soil temperature profiles, and that near surface temperature controls thaw depth.**

**This study provides some insight as to how mosses and other environmental factors influence soil microclimate and can influence permafrost thaw. However, I feel the current manuscript in general lacks clarity. It is not clear to me what exact knowledge gap this study is trying to fill and how the analysis used is addressing this. I believe the manuscript can be much improved by looking more into how these different drivers interact and the mechanisms involved. I will give further details below.**

We thank the referee for the detailed comments regarding how to improve our manuscript and what to consider before resubmission. We hope that, after addressing the comments of this and the other referee, our work and findings are communicated more clearly. In the revised manuscript we will more clearly specify the novelty of this work, and the unprecedented extent of field data collected for this study. A novelty in our results was highlighted by the reviewer as the lack of a species-specific effect on water retention seen in previous studies (e.g. Hrbáček et al. 2020). The methods and location of our study also provide a novelty. While some of the relationships explored in our study have been established in the past, they have not been studied in our region of study despite the heavy presence of moss nor have they collected datasets with data at a scale like ours. Studies such as Hrbáček et al., 2020; Gornall et al., 2007; and Porada et al., 2016 did not take separate measurements for the green and brown tissue (which our study can obtain by taking the difference between the total and green moss tissue thickness) but took one single measurement despite these two portions of the moss layer behaving differently and influencing active layer thaw in different ways. Our study also included a finer scale of moss mat thickness measurements which allowed us to observe breakpoints in our relationships which could not be explored with the limited measurements conducted in other studies. To our knowledge, past studies have depended on modeling with limited field data which have left gaps in our understanding of the full, quantifiable effect of the moss layer on active layer thaw. We will specify and better highlight these novelties in the revised manuscript.

**The introduction can be improved by restructuring the current text and better introducing the exact research question this study is addressing. At the moment, the introduction is a bit repetitive as certain points are made multiple times and arguments that belong together are spread across paragraphs. I also struggled to identify what knowledge gap this study is trying to fill and how the study addresses this. From the introduction it seems like the role of mosses in influencing soil temperature and thaw depth is already well established. I believe this study could still provide additional insight by looking more into specific mechanisms (non-linearity of responses), and interactions between different factors and context dependency.**

We will simplify the introduction to reduce repetition when preparing the manuscript for resubmission, and better specify the novelty of this study.

This study aimed to fill gaps in our understanding of the extent to which the moss layer reduces permafrost degradation. Our study aimed to accomplish this by providing a high-resolution set of quantitative data exploring previously understood but not thoroughly documented relationships. To our knowledge, previous research did not provide a dataset of the extent we presented here, and mostly focused on modelling with limited data for field validation.

As suggested by the reviewer, in the revised manuscript, we will specify mechanisms and interactions between different factors and better describe the complexity in these interactions.

**The Methods section is generally clear but needs a couple of clarifications, see specific comments.**

This will be addressed in the revised manuscript.

**The statistical analysis and result section lacks clarity. Many predictor variables are used, but there is little reflection as to why certain variables are used. I personally also think that there are too many graphs in the results section, which adds to me losing track of what the authors really are trying to show.**

This will be clarified when preparing the manuscript for resubmission. We will reduce the number of graphs and move some to supplementary information.

**The authors mention the correlation between variables and that these are not used together in multiple regression analysis. However, they do not provide clear information as to which variables were included in multiple regression and why.**

The variables selected for the multiple regression were those that did not present collinearity with each other. We will make sure to include a section that details what variables were selected and be clearer about why these were selected in the revised manuscript and include a table listing the variables collected.

**In the analysis, a number of temperature parameters are used, both as predictor and response variables. However, there is little explanation as to why these different parameters are of importance or why they may have different drivers. This lack of context makes it hard to see the relevance of these different analysis and because of the many different temperature variables it becomes hard to follow and confusing.**

We will make sure to add additional context to clarify the importance of the variables and why they were selected for the analyses in the revised manuscript.

**The authors use deviation form mean elevation (dz) as a predictor variable, which indicates that there are raised plots and hollows which have distinct environmental conditions and also influence moss thickness. However, the interrelatedness does not really come forward until later.**

We will make sure to discuss these concepts earlier in the revised manuscript.

**I also wonder whether the moss dominance varies between hollows and raised plots?**

We will test to see if there was a difference in moss dominance based on microtopography and report any significant differences in the revised manuscript.

**Hrbáček et al. 2020 found species-specific effect moss on ground surface temperature and active layer depth due to differences in water retention capacity and structure. The authors find no difference in the influence of moss genus, which to me is an interesting point and could be further discussed. Are the dominant species in your site very similar in structure and water retention capacity or could your results be confounded by topography.**

We will highlight these interesting results better and explore possible explanations for our findings in the revised manuscript.

**In my opinion, the analysis needs to better substantiated and should better reflect the interrelatedness of these different factors as illustrated by the visual summary. A structural equation model may be a more appropriate analysis which can show these dependencies. In a structural equation model air temperature, PAR and net radiation could be used as general climate conditions that influence soil temperature. Soil temperature can further be modulated by soil moisture content and moss thickness, which in concert regulate thaw depth. The present analysis is not able to tease apart the effect of moss layer thickness from environmental conditions and the resulting effect on thaw depth.**

We will  test if structural equation modelling is able to better explain the results reported in this study.

**The general readability of the manuscript can also be improved by splitting up long sentences.**

This will be addressed in the revised manuscript.

**Specific comments**

**Instead of a picture of the datalogger, a picture of the fiberglass probe would be more illustrative of the research method.**

This will be added to the revised manuscript.

**Line 110 Can the authors include specifics on the graminoid species present?**

We will include these details in the revised manuscript. Our team performed this species identification in a previous study (Davidson et al., 2016; https://www.researchgate.net/publication/303712736_Vegetation_Type_Dominates _the_Spatial_Variability_in_CH4_Emissions_Across_Multiple_Arctic_Tundra_Landscap es ).

**Line 120 states that moss and soil temperature were measured from 1cm below the surface? What do the authors mean with below the surface? Does it include temperatures in the moss layer as is alluded to? Or is it only soil temperatures starting from soil surface? This will matter for the analysis.**

Below the surface refers to any point below the green, photosynthetically active section of the moss layer and includes the organic soil layer composed of mineral soil and the brown, photosynthetically inactive section of the moss layer. We will specify this in the revised manuscript.

**Can the authors more clearly state the frequency of measurements of the different variables.**

Vegetation characteristics including the dominant genus, total moss layer thickness, and green layer thickness were collected once across the week of the 6[th] of July in 2021 (n = 124). During the week of the 8[th] of July in 2022, we conducted a random sampling (n = 20) of the plots to minimize the disturbance to the tundra. These random samples involved once again measuring the total moss layer thickness and green layer thickness and noting the dominant genus.

Belowground temperature data, thaw depth, water table level, and volumetric water content at the 124 plot points were collected on a weekly basis across both field seasons. Data was collected across 4 weeks in 2021 (July 8[th] – July 28[th]) and 4 weeks in 2022 (June 23[rd] – July 14[th]) for a total of 8 weeks of data. This will be clarified in the revised manuscript.

**Where measurements done in a randomized manner? Otherwise, one can expect diurnal patterns to influence the soil temperatures measured at specific locations.**

We attempted to collect these measurements consistently at similar times of the day. We will specify this in the revised manuscript.

**I suggest using a correlation matrix to illustrate the correlation between various parameters and the significance. This will provide a much clearer overview than the current supplementary table.**

We will include a correlation matrix in the revised manuscript.

**Taken together I believe this manuscript needs extensive revisions to provide clarity by clearly defining research questions and how these are addressed. I also believe the authors need to rethink their analysis, so it better matches the inter-relatedness of their predictors. I therefore suggest the authors to thoroughly revise this manuscript before a potential resubmission.**

We want to thank the referee once again for the detailed comments and hope that our revisions based on the comments provided will improve our manuscript's clarity. We feel that by including the reviewer's comments, the novelty of our study will be clearer, and considered appropriate for publication in Biogeosciences.

---

## Author Comment (AC3)

**The paper describes a study where the relationship between moss properties and environmental variables like thaw depth, soil temperature and moisture are investigated. The basis for the analysis is a dataset consisting of two years of measurements of these variables along transects in the Alaskan tundra. The authors run their dataset through various statistical models and find that**

**1. Moss layers drives larger temperature gradients in the ground and limits active layer thickness,**

**2. Surface temperatures are cooler in moister locations, and**

**3. That the topmost, live part of moss is important in regulating soil temperatures. The authors relate these findings to the low thermal conductivity of dry moss, the thermal conductivity and heat capacity of water, and the latent cooling associated with evaporation. The findings are briefly discussed.**

**After my reading of the paper, I find it to address a relevant research question, namely which factors regulate the energy transfer between the atmosphere and the ground in moss dominated ecosystems. While the topic of the paper is within the scope of Biogeosciences, I do not find it to present the novel and substantial scientific contribution that a publication in this journal necessitates. Most notably, the papers finding 1. and 2. are previously established relationships, and finding 3. is presenter without the appropriate discussion. Overall, I find that the paper has considerable shortcomings in several aspects including scientific significance, quality and presentation. I would thus recommend the authors to rewrite and resubmit the paper in an appropriate journal.**

We thank the referee for the considering our research question relevant and recommend avenues to improve the manuscript. We hope that after addressing these comments as well as those of the other referee, our manuscript will be considered appropriate for resubmission to this journal. The previous reviewer also highlighted how several interesting results of our study were not properly discussed and highlighted; we feel that by better developing the discussion and exploring additional statistical analysis, the novelty of our study will be clearer, and hope that our study will be considered appropriate for publication in Biogeosciences.

**The paper primarily investigates how soil temperatures and thaw depth are influenced by moss thickness and soil moisture, and the title should reflect that.**

This title change will be considered when revising the manuscript.

**The relevance, key aims, data sources and conclusions are presented in an orderly and appropriate way. As roughly ¼ of the introduction is about the risk of carbon release from permafrost soils, it would be appropriate to mention the processes through which mosses contributes to protecting this carbon stock. The vulnerability of mosses (Line 26-28) is also important context, but is not mentioned in the body of the paper (Introduction – conclusion).**

We will include a section in the abstract discussing the role of mosses in protecting the carbon stock as well as a section in the body of the paper on the vulnerability of mosses when revising the manuscript.

**The authors provide relevant background for their study, including the protected carbon stocks found in permafrost soils, positive feedback mechanisms upon permafrost degradation, and projected climate and wetness change. The paper however fails to revisit these important contexts in the discussion.**

This will be addressed in the revised manuscript, and the discussion will be further developed.

**A major issue in the introduction is that it is not made clear what research gap the authors investigate in the study. The goal is stated to be "to identify the biotic and abiotic controls regulating soil temperature and the thawing of the active layer", while the paragraph from Line 52-62 describes that there already is an established scientific understanding of the role of mosses in this context. The introduction should clearly state what gap in current scientific understanding the paper is addressing, and why their method and study site is appropriate. If the paper mostly aims to confirm well-known relationships, there needs to be an argument for why it is still relevant. For example, that the authors have a focus on previously underrepresented ecosystems/study areas/climates, or that the study is more quantitative than previous ones.**

We will better highlight the knowledge gaps in the current literature, and how our study addressed it. The other reviewer has highlighted this shortcoming as well and had useful suggestions.

**The study site description does not provide the reader with a proper overview of the site that is investigated. Here, I feel there should be a description of the general area in terms of climate, topography and dominant ecosystems. The local site description needs also to address the representativity of the site – i.e. can findings from this site be used elsewhere, both regionally and globally? The study site description also markets other data and experiments (eddy covariance, Biocomplexity etc.) without any clear explanation of why these are relevant to the paper. Credit of previous research efforts should be limited to the acknowledgements, whereas the study site description simply cites the studies that provide the data or statements used in the current study.**

We will include a more detailed description of the general area and the representativity of the site in the revised manuscript. We will clarify the relevance of other data, for example data from the eddy covariance tower were used in this manuscript.

**I also find the sampling description to be ambiguous. Why do the authors choose to sample along these transects? Do they follow some environmental or topographical gradients of interest? It is also completely unclear what the measurement period is, i.e. do you sample weekly year-round or only in a limited period in summer? And are the temperature and moisture measurements instantaneous or daily values? Without such metadata and general information, it is not possible for the reader to assess whether the acquired dataset is suitable to answer the questions at hand.**

Sampling was conducted along the transects because of the presence of the boardwalk mentioned in the manuscript; conducting our sampling on these boardwalks allowed us to minimize disturbances to the tundra. Data was collected across 4 weeks in 2021 (July 8th – July 28th) and 4 weeks in 2022 (June 23rd – July 14th) for a total of 8 weeks of data during what would be the growth period for vegetation in the area. Temperature and moisture measurements are instantaneous measurements. These details will be clarified in the revised manuscript.

**For the environmental variables, all those that are used in the study need to be listed, potentially in a table also indicating their units, annual range and type of sensor. From line 136-137 it is unclear if these are alle variables used in the study, or if there are additional ones not listed.**

We will include a table listing all the data collected and their intervals in the revised manuscript.

**This section (or the results) also lacks a short description of the data that is actually sampled. What are the annual ranges and averages? Was some sort of filtering applied? Are there clear clusters or thresholds for some of the variables?**

We will clarify this in the revised manuscript.

**I was surprised that the paper does not make an argument for the choice of statistical methods. While I am not an expert in statistical modelling, I would expect at least a simple statement to why the methods used in this study are suitable for the data and research question. This might be especially relevant as moisture has natural limits at 0 and 100%, and temperature has a hiatus around 0°C during thawing/freezing and thus does not behave linearly. I also found that stepwise regression is a controversial and partially discouraged method (e.g. Flom & Cassel (2007)), and I would expect the authors to comment on the applicability of this method.**

We will test additional statistical analyses as suggested by the other reviewer, and better explain our choices in the revised manuscript.

**It was also puzzling that in line 155 the depth -15 cm is stated to be "most consistently represented", while in 120-121 says temperature is recorded every centimetre until 20 cm for each plot. If there are issues implementing the sampling routine outlined, this should be mentioned and explained. In general, the state and nature of the dataset and choice of methods is not described in a manner fostering replicability.**

This will be clarified in the updated manuscript.

**This section does not present the findings in a clear and straight forward manner. Figures are presented before they appear in the text, there is a mixture of variable names and symbols in both text and graphs, and the metrics such as "Akaike information criterion" are not explained. It is also not**

**clear which results are based on some sort of average and which are time series (e.g. Figure 4 where soil water contents are regressed against maximum temperatures).**

We will organize the order of the graphs for clarity and improve the flow in the updated manuscript. We will clarify metrics and averages as we revise the manuscript.

**While the topics brought forward here are of relevance for the study, this section fails to convey how this study takes research forward. That mosses insulate the soil from the atmosphere and that evaporation cools the surface are established concepts within the field. I do find it highly interesting that the topmost live layer of moss has such a strong impact on soil temperatures, and would have expected a more thorough discussion of possible processes.**

We thank the reviewer for highlighting some interesting results in our manuscript, which will be more clearly discussed in the revised manuscript.

**A major issue with the paper is that the discussion does not revisit the important context provided in the introduction; the carbon stocks in permafrost soils, positive climate feedback mechanisms, changes in precipitation and the vulnerability of moss ecosystems. Several of these topics are strongly linked to the authors findings, and a proper discussion of them would greatly improve the relevance of the paper. The paper also needs to address the quality and robustness of the sampling routine, data and methods used, and potentially outline suggested improvements.**

We will expand our discussion to include undiscussed topics in the updated manuscript.

**This section is concise and to the point. The statements that a wider range of soil moisture (line 307) and moss thickness (line 308) would be required to understand this topic comes as a surprise as this topic is not mentioned in the discussion. The conclusion should briefly present the aims and how they were/where not achieved, rather than presenting new topics.**

These are very useful comments that we will include in the revised manuscript. We realized that the discussion should be better developed and the novelty of the results, and the relevance of the data collected should be better highlighted.

We thank the referee once again for the comments on the manuscript. We hope that by highlighting the novelty of our results, our study will be considered a significant contribution in this field of research.